# DIVERSITY OF TRANSFORMER LAYERS: ONE ASPECT OF PARAMETER SCALING LAWS

## ABSTRACT

Transformers deliver outstanding performance across a wide range of tasks and are now a dominant backbone architecture for large language models (LLMs). Their task-solving performance is improved by increasing parameter size, as shown in the recent studies on parameter scaling laws. Although recent mechanistic-interpretability studies have deepened our understanding of the internal behavior of Transformers by analyzing their residual stream, the relationship between these internal mechanisms and the parameter scaling laws remains unclear. To bridge this gap, we focus on layers and their size, which mainly decide the parameter size of Transformers. For this purpose, we first theoretically investigate the layers within the residual stream through a bias-diversity decomposition. The decomposition separates (i) bias, the error of each layer's output from the ground truth, and (ii) diversity, which indicates how much the outputs of each layer differ from each other. Analyzing Transformers under this theory reveals that performance improves when individual layers make predictions close to the correct answer and remain mutually diverse. We show that diversity becomes especially critical when individual layers' outputs are far from the ground truth. Finally, we introduce an information-theoretic diversity and show our main findings that adding layers enhances performance only when those layers behave differently, i.e., are diverse. We also reveal the performance gains from increasing the number of layers exhibit submodularity: marginal improvements diminish as additional layers increase, mirroring the logarithmic convergence predicted by the parameter scaling laws. Experiments on multiple semantic-understanding tasks with various LLMs empirically confirm the theoretical properties derived in this study. Our code will be available at `https://github.com/[Anonymous]`.

## 1 INTRODUCTION

Transformer (Vaswani et al., 2017) is one of the dominant architectures across a wide range of tasks in natural language processing (NLP) and other fields. In particular, large language models (LLMs) built on the Transformer backbone have demonstrated remarkable capabilities in language understanding and generation. This rapid progress, however, raises fundamental questions about why Transformers perform so well and how their performance scales with model size.

Empirical scaling studies have shown that increasing model parameter size leads to predictable improvements in performance (Hestness et al., 2017; Kaplan et al., 2020). These scaling laws suggest that larger networks consistently yield better performance as represented by in-context learning ability (Brown et al., 2020) and emergent capabilities observed in models with hundreds of billions of parameters (Wei et al., 2022).

Despite the impressive empirical successes of LLMs, their inner workings largely remain a black box. The mechanistic interpretability has partially elucidated the internal computations of Transformers and uncovered interpretable patterns and circuits within these networks (Geva et al., 2021; Olsson et al., 2022). Studies focusing on the residual stream (Elhage et al., 2021), composed of embedding, Multi-head Attention (MHA), and Multi-layer Perceptron (MLP) layers, have further deepened the interpretation of how information accumulates on residual networks through layers.

These works suggest that each layer contributes incrementally to the model's prediction by adding or refining information in the residual stream. However, the relationship between these internal

Figure 1: The overview of the residual stream in pre-layer normalization type Transformers.

layer-wise mechanisms and the observed scaling behavior of the overall model remains unclear. This disconnect limits the efficient performance improvement of Transformers.

To address this gap, we propose a theoretical interpretation that links a Transformer's internal layer dynamics with its overall performance, offering a new lens to interpret parameter scaling laws. As a first step, we rely on bias–diversity decomposition (Krogh & Vedelsby, 1994) to quantify the contribution of each layer within the residual stream. In this formulation, each layer's output is evaluated in terms of (i) bias, the error between the layer's output prediction and the ground-truth target, and (ii) diversity, showing how layers' outputs differ from each other. By analyzing Transformers under this approach, we obtain theoretical insights that achieving both low bias and high diversity across layers is ideal for improving performance. In contrast, bias and diversity depend on each other, which makes it difficult to improve both bias and diversity.

Finally, to deal with our main target, parameter scaling laws, from an information-theoretic diversity (Brown, 2009; Zhou & Li, 2010), we examined how diversity contributes to performance gains when additional layers are introduced. Our analysis shows that, to improve performance, the outputs of different layers must remain distinct, i.e., diverse. We also provide a theoretical explanation for the diminishing returns of adding layers: the marginal performance improvement decreases as more layers are stacked. This tendency is consistent with the Scaling Law, which predicts that performance grows logarithmically with the number of parameters.

Finally, we conduct experiments on multiple NLP tasks using various LLMs and show that our theoretical findings are valid in practical situations.

## 2 RESIDUAL STREAM IN TRANSFORMERS

As shown in Figure 1, in pre-layer normalization type transformers (Xiong et al., 2020) commonly used in LLMs, there is a residual stream (Elhage et al., 2021) consisting of embedding layers, multi-head attention (MHA), and multi-layer perceptron (MLP). In this residual stream, the outputs of each module are added together, and finally, the normalized result is projected to predict the output. Letting $L = (\mathbf{z}^{(1)}, \cdots, \mathbf{z}^{(|L|)})$, $E = (\mathbf{e}^{(1)}, \cdots, \mathbf{e}^{(|E|)})$, $M = (\mathbf{m}^{(1)}, \cdots, \mathbf{m}^{(|M|)})$, and $A = (\mathbf{a}^{(1)}, \cdots, \mathbf{a}^{(|A|)})$ be sequences of entire layers, embedding layers, MHA layers, and MLP layers, respectively, the logit prediction on the residual stream is represented as follows:

$$\mathbf{logits} = W_{out}\mathrm{LN}(\sum_{i=1}^{|L|}\mathbf{z}^{(i)}) = W_{out}\mathrm{LN}(\sum_{i=1}^{|E|}\mathbf{e}^{(i)} + \sum_{i=1}^{|M|}\mathbf{m}^{(i)} + \sum_{i=1}^{|A|}\mathbf{a}^{(i)}), \quad (1)$$

where LN is RMSNorm (Zhang & Sennrich, 2019) and $W_{out}$ is a projection layer. By representing the scaling by RMSNorm as $\mathbf{s}$, we can reformulate Eq. 1 as follows (Chang et al., 2024):

$$(1) \equiv \mathbf{logits} = W_{out}(\mathbf{s} \odot \sum_{i=1}^{|L|}\mathbf{z}^{(i)}) = \sum_{i=1}^{|L|} W_{out}.(\mathbf{s} \odot \mathbf{z}^{(i)}) \quad (2)$$

Here, by replacing $W_{out}(\mathbf{s} \odot \mathbf{z}^{(i)})$ with $\mathbf{u}^{(i)}$, we can reformulate Eq. 2 as follows:

$$\mathbf{logits} = \sum_{i=1}^{|L|}\mathbf{u}^{(i)}. \quad (3)$$

We use this equation to investigate the theoretical effects of modules in the residual stream.

## 3 THEORETICAL ANALYSIS

In this section, we analyze the theoretical characteristics of the residual stream of Transformer layers.

### 3.1 PREDICTION DISCREPANCY

In general, Transformers output the token with the highest prediction probability for a given input. This behavior can be formalized as follows:

$$\arg\max_{y} \mathbf{logits} = \arg\max_{y} \sum_{i=1}^{|L|} \mathbf{u}^{(i)} = \arg\max_{y} \frac{1}{|L|} \sum_{i=1}^{|L|} \mathbf{u}^{(i)} \tag{4}$$

Here, we can reformulate Eq. 4 with $\bar{\mathbf{u}} = \frac{1}{|L|} \sum_{i=1}^{|L|} \mathbf{u}^{(i)}$ as follows:

$$(4) \equiv \arg\max_{y} \bar{\mathbf{u}} \tag{5}$$

By defining $\hat{\mathbf{u}}$ as the true distribution of logits $\bar{\mathbf{u}}$, we can consider a prediction error between Transformer layers and its true distribution on logits as the following Mean Squared Error (MSE):

$$MSE(\hat{\mathbf{u}}, \bar{\mathbf{u}}) = \frac{1}{|V|} \sum_{j=1}^{|V|} (\hat{u}_j - \bar{u}_j)^2 = \mathbb{E}_{j \in V}[(\hat{u}_j - \bar{u}_j)^2]. \tag{6}$$

We use this error to theoretically analyze the discrepancy between the predicted logits by the Transformer and the true logit distribution to reveal the theoretical insights into the residual streams of the Transformer layers.

### 3.2 IMPORTANCE OF DIVERSITY

First, we focus on the diversity arising from the differences between the predictions of each Transformer layer. From Eqs. 1 and 6, we can decompose the discrepancy between the Transformer's predictions and the true logit distribution into bias and diversity terms as in the following theorem.

**Theorem 1 (Bias and Diversity Decomposition).** *The prediction error of Transformer layers, $MSE(\hat{\boldsymbol{u}}, \bar{\boldsymbol{u}})$, can be decomposed into bias and diversity (ambiguity) (Wood et al., 2024) terms (Krogh & Vedelsby, 1994) as follows:*

$$MSE(\hat{\boldsymbol{u}}, \bar{\boldsymbol{u}}) \tag{7}$$

$$= \underbrace{\mathbb{E}_{j \in V}[\mathbb{E}_{\mathbf{u} \in L}[(\hat{u}_j - u_j)^2]]}_{Bias} - \underbrace{\mathbb{E}_{j \in V}[\mathbb{E}_{\mathbf{u} \in L}[(\bar{u}_j - u_j)^2]]}_{Diversity} \tag{8}$$

$$= \underbrace{\frac{1}{|L|}\mathbb{E}_{j \in V}[|E|\underbrace{\mathbb{E}_{\mathbf{e} \in E}[(\hat{u}_j - e_j)^2]}_{Embedding\ Bias} + |M|\underbrace{\mathbb{E}_{\mathbf{m} \in M}[(\hat{u}_j - m_j)^2]}_{MLP\ Bias} + |A|\underbrace{\mathbb{E}_{\mathbf{a} \in A}[(\hat{u}_j - a_j)^2]}_{Attention\ Bias}}_{Bias} \tag{9}$$

$$- \underbrace{\frac{1}{|L|}\mathbb{E}_{j \in V}[|E|\underbrace{\mathbb{E}_{\mathbf{e} \in E}[(\bar{u}_j - e_j)^2]}_{Embedding\ Diversity} + |M|\underbrace{\mathbb{E}_{\mathbf{m} \in M}[(\bar{u}_j - m_j)^2]}_{MLP\ Diversity} + |A|\underbrace{\mathbb{E}_{\mathbf{a} \in A}[(\bar{u}_j - a_j)^2]}_{Attention\ Diversity}}_{Diversity}. \tag{10}$$

*(See Appendix A.1 for the proof.)*

As shown in Eq. 8, MSE can be decomposed into two terms: bias and diversity. Since the diversity term is negative, we can see that when diversity increases, the discrepancy from the true distribution decreases, meaning that prediction accuracy improves. This is different from the bias-variance decomposition, which is widely known in the field of machine learning.

Equations 9 and 10 show the further decomposition of the bias and diversity terms corresponding to the modules on the residual stream in Eq. 1. From this, we can see that each module contributes to both bias and diversity, and that the number of modules weighs the contribution. Therefore, we

can confirm that MLP layers and attention layers play an important role in Transformers in terms of the number of layers. The contribution of the embedding layer is small based on the number of layers, while in contrast, its relative importance may increase in lightweight models. We empirically investigate these hypotheses through experiments in §5.3.

### 3.3 Limitation of Diversity

As discussed in §3.2, the diversity term is important to improve the prediction performance. However, there are limitations to its effectiveness. The next theorem indicates the limitation.

**Theorem 2** (**Limitation of Diversity**). *The decomposition of the prediction error in Transformer layers (Eqs. 8, 9, and 10) holds the following relation: $Diversity \rightarrow 0$ ($Bias \rightarrow 0$); $Embedding\ Diversity \rightarrow 0$ ($Embedding\ Bias \rightarrow 0$); $MLP\ Diversity \rightarrow 0$ ($MLP\ Bias \rightarrow 0$); $Attention\ Diversity \rightarrow 0$ ($Attention\ Bias \rightarrow 0$). (See Appendix A.2 for the proof.)*

From Theorem 2, regarding the prediction by entire modules and each module, when bias is close to zero, we cannot expect performance improvement from diversity. Therefore, when the predictions of each layer are sufficiently accurate, performance improvement by increasing diversity is limited. However, this relationship does not guarantee that diversity will increase when bias is far from zero. Therefore, when the predictions of each layer are inaccurate, the importance of diversity increases. Even under this situation, the following limitation exists.

**Theorem 3** (**Bias and Diversity Trade-off**). *Bias and Diversity in the decomposition of the prediction error in Transformer layers (Eq. 8) depend on each other as follows (Brown et al., 2005):*

$$Bias = \overline{bias}^2 + \Omega,\ Diversity = \Omega - \left[\frac{1}{|L|}\overline{var} + \left(1 - \frac{1}{|L|}\right)\overline{cov}\right], \qquad (11)$$

*where $\overline{bias} = \mathbb{E}_{\mathbf{u} \in L}[\mathbb{E}_{j \in V}[u_j] - \mathbb{E}_{j \in V}[\hat{u}_j]]$, $\overline{var} = \mathbb{E}_{\mathbf{u} \in L}[\mathbb{E}_{j \in V}(u_j - \mathbb{E}_{k \in V}[u_k])^2]$, $\overline{cov} = \frac{1}{|L|(|L|-1)}\sum_{i=1}^{|L|}\sum_{i \neq j}^{|L|}\mathbb{E}_{k \in V}[(u_k^i - \mathbb{E}_{l \in V}[u_l^i])(u_k^j - \mathbb{E}_{l \in V}[u_l^j])]$, and $\Omega = \overline{var} + \mathbb{E}_{\mathbf{u} \in L}[(\mathbb{E}_{j \in V}[u_j] - \mathbb{E}_{j \in V}[\overline{u}_j])^2]$ (See Appendix A.3 for the proof.)*

By focusing on Eq. 11, we can see that the bias and diversity terms share $\Omega$. This suggests that attempting to reduce the bias term may also reduce $\Omega$, which in turn may reduce the diversity term. In other words, Theorem 3 demonstrates the general trade-off relationship between the bias and diversity term, emphasizing the difficulty of maximizing the diversity term without affecting the bias term.

### 3.4 Limitation of Stacking Layers

In §3.3, we discussed a trade-off relationship between the bias and diversity terms, and improving the diversity term can potentially improve the prediction performance of Transformers when the bias term is large. A simple method to improve performance is to add layers following the parameter scaling laws, and it can also improve the diversity based on our analysis so far. We theoretically investigate the performance improvement and limitations of this method through the following theorems.

**Theorem 4** (**Generalized Diversity**). *Letting $Y$ be a true label, variables $\mathbf{u}^{(1)}, \cdots, \mathbf{u}^{(|L|)}$ be $U_{1:|L|}$, and $g(U_{1:|L|})$ be a function[1] predicting the best label. Its prediction error $p(Y \neq g(U_{1:|L|}))$ satisfies the following inequality (Brown, 2009; Zhou & Li, 2010):*

$$\frac{H(Y) - I(U_{1:|L|}; Y) - 1}{\log |Y|} \leq p(Y \neq g(U_{1:|L|})) \leq \frac{H(Y) - I(U_{1:|L|}; Y)}{2}, \qquad (12)$$

---

[1]This function must follow the Bayes decision rule according to Hellman & Raviv (1970). In the case of the Transformer, the Bayesian decision rule stands only when the output distribution of the Transformer fits that of its training data, and argmax is used for its prediction. Since we target scaling laws that require large parameters with large data, this premise is valid and natural. Note that even a one-layer Transformer can satisfy the Bayesian decision rule (Shen et al., 2025; Nguyen & Nguyen-Tang, 2025).

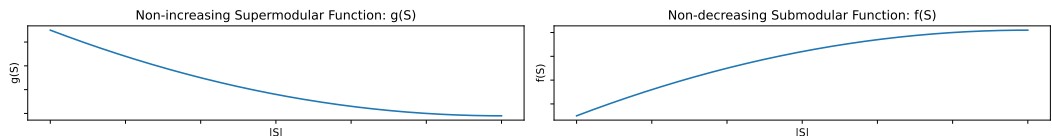

Figure 2: The examples of non-increasing supermodular and non-decreasing submodular functions.

where $H(Y)$ is entropy and $I(U_{1:|L|}; Y)$ is joint mutual information. To minimize the error, we should maximize $I(U_{1:|L|}; Y)$ which can be decomposed to (Zhou & Li, 2010):

$$I(U_{1:|L|}; Y) = \underbrace{\sum_{i=1}^{|L|} I(\mathbf{u}^{(i)}; Y)}_{Relevancy} + \underbrace{\underbrace{\mathcal{I}(U_{1:|L|}|Y)}_{Conditional\ Redundancy} - \underbrace{\mathcal{I}(U_{1:|L|})}_{Redundancy}}_{Information\ Theoretic\ Diveristy}, \tag{13}$$

where $\mathcal{I}(U_{1:|L|})$ is total correlation and $\mathcal{I}(U_{1:|L|}; Y)$ is conditional total correlation. (See Appendix A.4 for the proof.)

We can interpret Theorem 4 from the bias and diversity decomposition in Eq. 10 from a viewpoint of information theory, indicating that the bias term corresponds to the relevancy term and the diversity term corresponds to the information-theoretic diversity. Through this decomposition, we discuss how adding layers improves the prediction performance of Transformers.

**Theorem 5** (**Monotonicity of each Term**). *In Eq. 13, when $U_{1:|L|}$ increases (that means new elements are added to $U_{1:|L|}$), Relevancy, Conditional Redundancy, and Redundancy monotonically increase, whereas Information Theoretic Diveristy and $I(U_{1:|L|}; Y)$ do not monotonically increase or decrease (See Appendix A.5 for the proof.)*

Theorem 5 suggests that simply adding more layers to a Transformer does not guarantee an improvement in performance, especially due to the difficulty of improving information-theoretic diversity. This insight is further reinforced by the following theorem.

**Theorem 6** (**Monotonicity of Upper and Lower Bounds**). *In Eq. 12, the lower bound $\frac{H(Y) - I(U_{1:|L|}; Y) - 1}{\log |Y|}$ and the upper bound $\frac{H(Y) - I(U_{1:|L|}; Y)}{2}$ do not monotonically increase or decrease when $U_{1:|L|}$ increases. (See Appendix A.6 for the proof.)*

These results seem to indicate that increasing the number of layers in Transformers makes it difficult to improve performance, but at the same time, they suggest that performance can be improved by enhancing diversity corresponding to *Information Theoretic Diveristy*. Therefore, similar to Theorem 1, diversity is also important from an information-theoretical perspective. However, regarding the information-theoretic diversity, there is a limitation described by the following theorem.

**Theorem 7** (**Submodularity of each Term**). *In Eq. 13, when the variables in $U_{1:|L|}$ are independent given $Y$, $I(U_{1:|L|}; Y)$ and Information Theoretic Diversity are submodular, and $I(U_{1:|L|}; Y)$ is non-decreasing on $U_{1:|L|}$. (See Appendix A.7 for the proof.)*

This theorem assumes a kind of lower bound that *Conditional Redundancy* becomes zero due to the variables in $U_{1:|L|}$ being independent given $Y$. Even in this situation, the performance of Transformers increases by adding layers because $I(U_{1:|L|}; Y)$ is non-decreasing. Here, submodularity refers to the property whereby an increase in input leads to decreasing additional benefit in output. Thus, the effect of adding layers on performance decreases as the number of layers increases in this situation. By assuming that the number of layers and the number of parameters are proportional, this result is consistent with the parameter scaling laws, which show that performance converges logarithmically with an increase in the number of parameters. The next theorem reinforces this characteristic.

**Theorem 8** (**Supermodularity of Upper and Lower Bounds**). *In Eq. 12, when the variables in $U_{1:|L|}$ are independent given $Y$, the lower bound $\frac{H(Y) - I(U_{1:|L|}; Y) - 1}{\log |Y|}$ and the upper bound $\frac{H(Y) - I(U_{1:|L|}; Y)}{2}$ are supermodular and non-increasing on $U_{1:|L|}$. (See Appendix A.8 for the proof.)*

Supermodularity is a property that the more inputs there are, the greater the risk that additional inputs will result in performance not improving. Therefore, this theorem is also consistent with the parameter scaling laws and reinforces Theorem 7.

Figure 2 shows the example behaviors of non-increasing supermodular and non-decreasing submodular functions. The gradual convergence of values aligns with the logarithmic convergence of parameter scaling laws. However, these are based on a kind of lower bound that *Conditional Redundancy* becomes zero. In the actual situation, we can expect performance improvement by increased *Conditional Redundancy*, and it makes the performance improvement non-monotonic, by following Theorems 5 and 6. We check the actual behavior in §5.5.

## 4 REMAINING PROBLEMS

### 4.1 DIFFERENCE BETWEEN PROBABILITY AND LOGITS

Previous discussions have focused on the logits in the output layer of the Transformer. However, when actually using the Transformer, the output is selected from vectors normalized by the softmax layer. Therefore, to investigate whether our theoretical findings are practical, it is desirable to conduct experiments using actual models and datasets. To investigate the behavior of MSE and bias through experiments, we should prepare the true logit distribution, $\hat{\mathbf{u}}$. However, assuming the correct answers in softmax, softmax returns the same value for shifts in the input like $\frac{e^{\mathbf{logits}_i}}{\sum_j e^{\mathbf{logits}_j}} = \frac{e^{(\mathbf{logits}_i - \mu)}}{\sum_j e^{(\mathbf{logits}_j - \mu)}}$, resulting in an infinite number of true logits, which is difficult to handle. To address this issue, we approximate MSE and bias in Eq. 8 using the following equation:

$$MSE(\hat{\mathbf{u}}, \bar{\mathbf{u}}) \approx MSE(\tilde{\mathbf{u}}, softmax(\bar{\mathbf{u}})); bias \approx \frac{1}{|V|} \sum_{i=1}^{|V|} \frac{1}{|L|} \sum_{j=1}^{|L|} \left( \tilde{u}_i - softmax(\mathbf{u}^{(j)})_i \right)^2, \quad (14)$$

where $\tilde{\mathbf{u}}$ represents a one-hot vector whose dimension of a gold label is one and others are zero.

### 4.2 OUR ASSUMPTION FOR THE PARAMETER SCALING LAWS.

**Dimension Size.** Our previous discussion on the parameter scaling laws has been based on the assumption that the number of layers in a Transformer is proportional to the number of parameters. However, this assumption overlooks several important factors. One such factor is the number of dimensions. In many models, the actual number of parameters is represented by the product of the number of dimensions and the number of layers. Therefore, when considering the relationship with the parameter scaling laws in a rigorous manner, it is necessary to take into account elements at the neuron level corresponding to the dimension size. Fortunately, it is known that the residual stream in Transformers can be decomposed down to the neuron level (Elhage et al., 2021), allowing us to replace "layers" with "neurons" in our discussion and maintain the validity of our claims.

**Reuse of Layers.** Another oversight is the existence of models that recursively utilize layers. In such cases, the number of layers does not necessarily correspond to the number of parameters. A well-known example of such a model is the Universal Transformer (Dehghani et al., 2019), whose efficiency is also recognized in natural language generation tasks (Takase & Kiyono, 2023). Recently, there has been a trend to repurpose such models as small-scale LLMs (Touvron et al., 2023). We verify whether the increase in layers independent of parameters in such models is consistent with our claim in §5.6.

## 5 EMPIRICAL ANALYSIS

### 5.1 GENERAL SETTINGS

We used the following Transformer models from HuggingFace Transformers (Wolf et al., 2020): facebook/MobileLLM-125M; facebook/MobileLLM-125M-layer-share; facebook/MobileLLM-350M; facebook/MobileLLM-350M-layer-share (Liu et al., 2024); meta-llama/Llama-2-7b-hf;

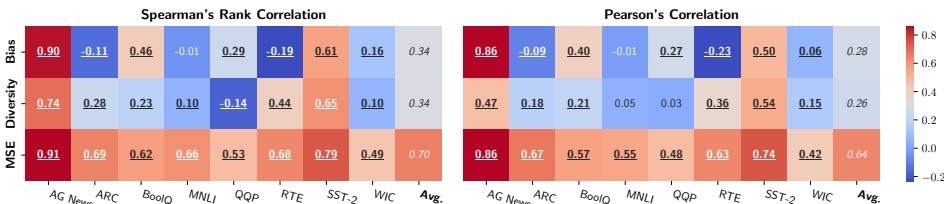

Figure 3: The correlation between accuracy and MSE, bias, and diversity of models across tasks. The underlined scores show the statistical significance ($p < 0.05$).[3] Note that the scores on Avg. are not the target of the significance test.

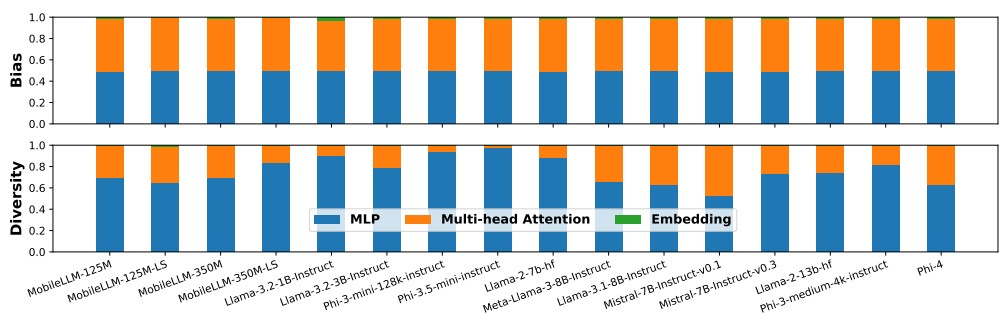

Figure 4: The average proportion of bias and diversity accounted for by each module in each model in all datasets.

meta-llama/Llama-2-13b-hf (Touvron et al., 2023); meta-llama/Meta-Llama-3-8B-Instruct; meta-llama/Llama-3.1-8B-Instruct; meta-llama/Llama-3.2-1B-Instruct; meta-llama/Llama-3.2-3B-Instruct Grattafiori et al. (2024); microsoft/Phi-3-medium-4k-instruct; microsoft/Phi-3-mini-128k-instruct; microsoft/Phi-3.5-mini-instruct Abdin et al. (2024a); microsoft/phi-4 (Abdin et al., 2024b); mistralai/Mistral-7B-Instruct-v0.1; mistralai/Mistral-7B-Instruct-v0.3 (Jiang et al., 2023).

By following the setting of the previous work (Chang et al., 2024), we used the first 2000 instances in AG News (Zhang et al., 2015), ARC (Clark et al., 2018), BoolQ (Clark et al., 2019), MNLI (Williams et al., 2018), QQP (Wang et al., 2017), RTE, SST-2, and WIC (Wang et al., 2019) datasets. We use the accuracy as a metric for each task. We used a predefined template by for prompting on each task and restricted the output to predefined options (See Appendix B for the details).

## 5.2 CORRELATION OF MSE, BIAS, AND DIVERSITY TO PERFORMANCE

In order to clarify the relationship between MSE, bias, diversity, and performance in actual tasks, we calculated the Pearson and Spearman's correlation between the accuracy and MSE, bias, and diversity at each junction point on the residual stream. Figure 3 shows the correlation.[4] As can be seen from these results, MSE shows a high correlation with accuracy. On the other hand, bias and diversity show moderate correlations, but the trends differ depending on the task, indicating that the model changes the roles of each layer according to the task. Furthermore, the fact that the Spearman correlation is lower than the rank correlation indicates that detailed accuracy differences cannot be read from MSE.

---

[3]This is based on Student's t-test (Student, 1908).

[4]Different from accuracy and diversity, lower bias and MSE are better for performance improvement. To reflect it, we used the negative value of bias and MSE when calculating the correlation. Also, to assign the value range, we standardize each score for each model. By utilizing z-transformation (Corey et al., 1998), we report averaged correlation (Avg.).

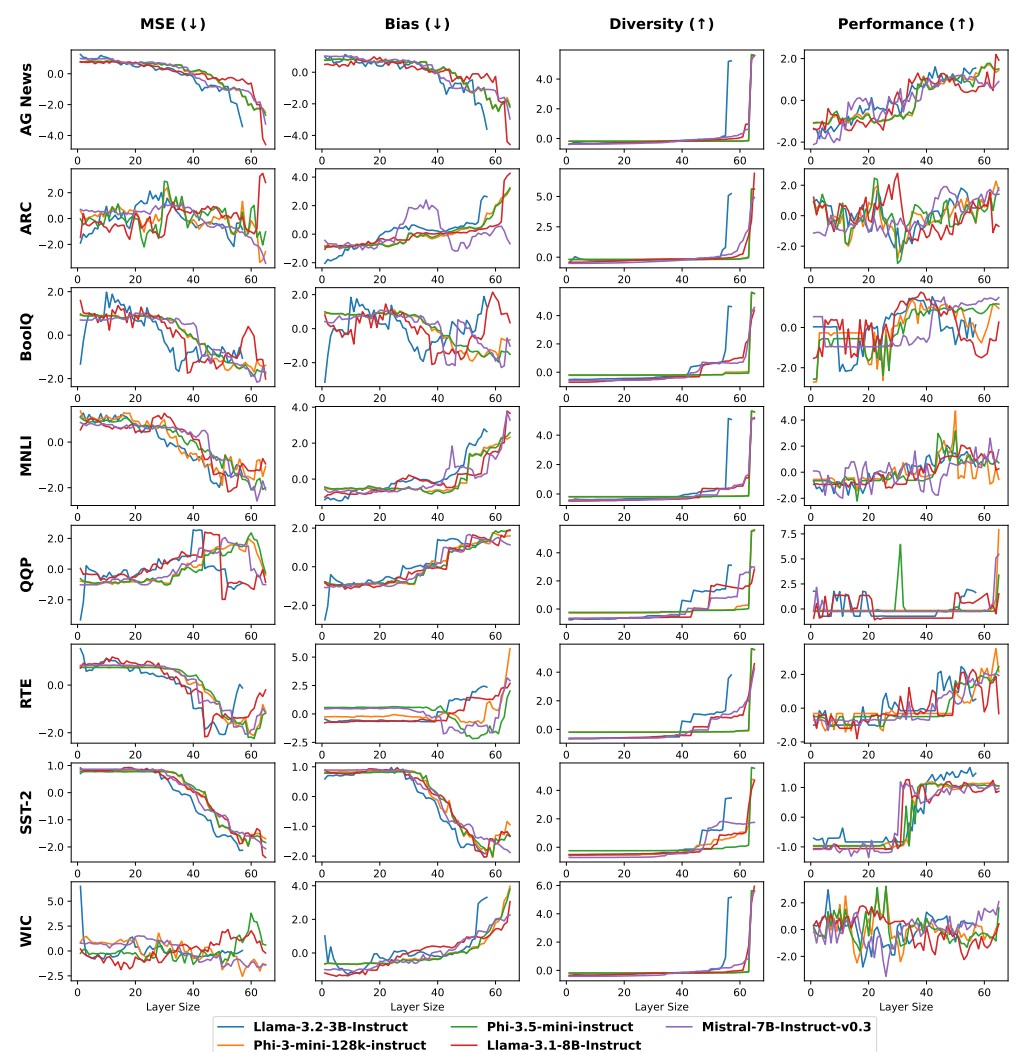

Figure 5: The relationship between MSE, bias, diversity, and performance on each task. Note that each metric is standardized to capture the change by the number of layers.

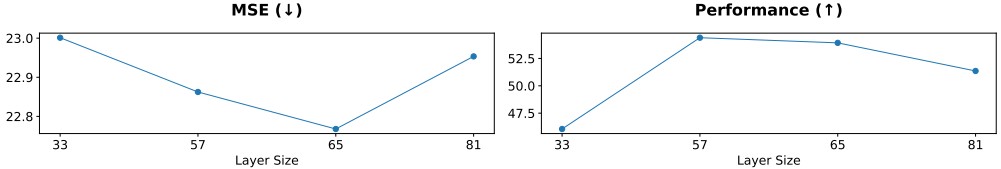

Figure 6: The relationship between performance and the number of layers.

## 5.3 INFLUENCE OF EACH MODULE

Figure 4 shows the influence of each module on bias and diversity based on the weights corresponding to their layer size, represented by Eqs. 9 and 10. In bias, the contributions for each module are proportional to their layer size. This is the reason for the few contributions of the embedding layers. These tendencies are almost the same across the models. In contrast, the MLP layers dominate diversity. However, the tendency differs depending on the model. This result suggests the necessity of improving the diversity of multi-head attention layers. It supports the previous work, indicating

Table 1: The results of MobileLLM on different tasks. Avg. indicates the average scores of all tasks. The suffix -LS indicates that the model recurrently shares its layers.

| MobileLLM | Accuracy | | | | | | | | | MSE | | | | | | | | |
|---|---|---|---|---|---|---|---|---|---|---|---|---|---|---|---|---|---|---|
| | AG News | ARC | BoolQ | MNLI | QQP | RTE | SST-2 | WIC | Avg. | AG News | ARC | BoolQ | MNLI | QQP | RTE | SST-2 | WIC | Avg. |
| 125M | 49.4 | 27.2 | 62.6 | 30.4 | 43.8 | 50.9 | 53.1 | 53.0 | 46.3 | 18.5 | 18.7 | 24.9 | 22.3 | 25.0 | 25.0 | 25.0 | 25.0 | 23.0 |
| 125M-LS | 47.9 | 28.4 | 62.7 | 30.4 | 34.8 | 52.7 | 55.4 | 52.7 | 45.6 | 18.7 | 18.7 | 25.0 | 22.2 | 25.0 | 25.0 | 25.0 | 25.0 | 23.1 |
| 350M | 50.4 | 27.7 | 62.6 | 30.8 | 35.2 | 53.4 | 52.6 | 55.2 | 46.0 | 18.5 | 18.7 | 24.9 | 22.2 | 25.0 | 25.0 | 25.0 | 25.0 | 23.0 |
| 350M-LS | 52.4 | 25.6 | 59.7 | 30.4 | 34.6 | 52.7 | 49.1 | 50.0 | 44.3 | 18.7 | 18.8 | 25.0 | 22.2 | 25.3 | 25.0 | 25.0 | 25.0 | 23.1 |

| MobileLLM | Bias | | | | | | | | | Diversity | | | | | | | | |
|---|---|---|---|---|---|---|---|---|---|---|---|---|---|---|---|---|---|---|
| | AG News | ARC | BoolQ | MNLI | QQP | RTE | SST-2 | WIC | Avg. | AG News | ARC | BoolQ | MNLI | QQP | RTE | SST-2 | WIC | Avg. |
| 125M | 18.6 | 18.8 | 25.1 | 22.4 | 25.7 | 25.2 | 25.0 | 25.3 | 23.3 | 6.2 | 10.1 | 11.8 | 8.1 | 20.3 | 6.2 | 21.8 | 13.1 | 12.2 |
| 125M-LS | 18.7 | 18.8 | 25.0 | 22.3 | 25.2 | 25.1 | 25.0 | 25.0 | 23.1 | 1.3 | 1.6 | 3.0 | 0.8 | 2.3 | 1.5 | 2.6 | 1.1 | 1.8 |
| 350M | 18.6 | 18.8 | 25.0 | 22.3 | 25.5 | 25.1 | 25.0 | 25.2 | 23.2 | 9.6 | 9.2 | 16.7 | 8.0 | 3.9 | 10.4 | 54.0 | 12.1 | 15.5 |
| 350M-LS | 18.7 | 18.8 | 25.1 | 22.3 | 25.5 | 25.1 | 25.1 | 25.1 | 23.2 | 2.9 | 4.6 | 8.6 | 3.2 | 2.6 | 3.0 | 5.7 | 3.4 | 4.2 |

the existence of unnecessary attention heads in Transformers (He et al., 2024). Similar to the case of the bias, the contribution of embedding layers is limited.

## 5.4 TRADE-OFF RELATIONSHIP BETWEEN BIAS AND DIVERSITY

We investigate whether there is actually a trade-off between bias and diversity, as shown in Theorems 2 and 3. Figure 5 shows the results.[5] As can be seen from these results, when there is no performance improvement, there is a trade-off between bias and diversity, but when performance improves, the trade-off relationship does not hold. This is consistent with the fact that bias and diversity share only one term, as shown in Theorem 3. The limitation of diversity shown in Theorem 2 is caused by bias being close to zero, but in reality, bias rarely approaches zero, so the decrease in bias does not hinder the improvement of diversity. Therefore, although a trade-off relationship may exist in some cases, it is possible to improve both bias and diversity.

## 5.5 DECREASE IN EFFICIENCY WITH INCREASING NUMBER OF LAYERS

We verify whether the decrease in performance improvement accompanying the increase in layers shown in Theorems 5, 6, 7, and 8 actually occurs. To do so, we averaged the accuracy and MSE for all tasks in models with no less than 1B of parameters. Figure 6 shows the result. From these results, we can see that the effectiveness of increasing layers actually decreases. Also, the performance degradation caused by increasing layers indicates that *Conditional Redundancy* becomes larger than zero, and the layers depend on each other to predict the answers.

## 5.6 CONSISTENCY OF OUR THEORY WITH MODELS THAT REUSE LAYERS

We verify the performance of a model that reuses layers. Table 1 shows the results obtained using MobileLLM. Basically, we find that the performance improvement achieved by reusing layers is limited. Furthermore, since the diversity of models that reuse layers is generally low, we consider that the low diversity is the reason for the limited performance improvement. This suggests that in order to improve model performance by reusing layers, it is necessary to improve the diversity of each module.

## 6 CONCLUSION

In this paper, we demonstrated that the diversity of predictions from each module on the residual stream of Transformers is important for improving performance from the perspective of bias and diversity decomposition. Additionally, we demonstrated limits to the performance improvements achievable through this diversity. Furthermore, we showed that performance improvements achieved by adding more layers are related to this diversity and that the effectiveness of these improvements diminishes as the number of layers increases, which is consistent with the parameter scaling laws. Experimental results across multiple tasks with various LLMs confirmed that empirical observations support these theoretical claims.

---

[5]Note that due to low visibility, this figure only covers models that achieved the top five accuracy on average over all tasks. See Appendix C for the overall result.

ETHICS STATEMENT

Because our work uses predefined output and templates for the investigation, there is no possibility of generating harmful content from LLMs. Furthermore, our work can contribute to revealing the internal behavior of LLMs to understand given instructions, which is often useful to prevent LLMs from generating harmful content.

REPRODUCIBILITY STATEMENT

Experimental settings and their details are described in §5 and Appendix B, respectively. Both sections cover the essential information for reproducing our reported results, which are about the used templates, models, decoding methods, and GPU environment. In addition, we will release our code for reproducibility.

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

# A    PROOFS

## A.1    THE PROOF FOR THEOREM 1

According to Krogh & Vedelsby (1994), when the coefficients of each term in Eq. 6 satisfy the definition of probability, we can decompose Eq. 6 into Eq. 8. Here, the coefficients follow a uniform distribution, and the decomposition of Eq. 8 holds. We can induce Eqs. 9 and 10 based on the fact that the weighted mean equals the original mean.

## A.2    THE PROOF FOR THEOREM 2

When the bias term in Eq. 8 equals zero, $\hat{u} = u$ holds. In this condition, $\bar{u} = \sum u$ becomes $\hat{u}$. Thus, the diversity terms become zero. Similarly, other decomposed diversity terms become zero when corresponding bias terms become zero.

## A.3    THE PROOF FOR THEOREM 3

The decomposition of Krogh & Vedelsby (1994) is a sufficient condition for the decomposition of Brown et al. (2005). Since we proved that Theorem 2 holds, Theorem 3 also holds.

## A.4    THE PROOF FOR THEOREM 4

See Zhou & Li (2010) for the proof of the bound and decomposition. Since their claim does not restrict the elements of the input and output in their formulation, we can apply their bound and decomposition to the behavior of the residual stream in Transformers.

## A.5    THE PROOF FOR THEOREM 5

Since *Relevancy* is a sum of joint mutual information, *Relevancy* increases when $|U_{1:|L|}|$ increases. This is because joint mutual information always becomes a positive value.

*Conditional Redundancy* is conditional total correlation (conditional multi-information). We can consider its increase when $U_{1:|L|}$ obtains a new element, $\mathbf{u}^{(|L|+1)}$ as follows:

$$\mathcal{I}(U_{1:|L|+1}|Y) - \mathcal{I}(U_{1:|L|}|Y) \tag{15}$$

$$= \left( \sum_{i=1}^{|L|+1} H(\mathbf{u}^{(i)}|Y) \right) - H(U_{1:|L|+1}|Y) - \left( \sum_{i=1}^{|L|} H(\mathbf{u}^{(i)}|Y) \right) + H(U_{1:|L|}|Y) \tag{16}$$

$$= H(\mathbf{u}^{(|L|+1)}|Y) - H(U_{1:|L|+1}|Y) + H(U_{1:|L|}|Y) \tag{17}$$

$$= H(\mathbf{u}^{(|L|+1)}|Y) - \left( H(\mathbf{u}^{(|L|+1)}|Y) + H(U_{1:|L|}|Y, \mathbf{u}^{(|L|+1)}) \right) + H(U_{1:|L|}|Y) \tag{18}$$

$$= H(U_{1:|L|}|Y) - H(U_{1:|L|}|Y, \mathbf{u}^{(|L|+1)}) \tag{19}$$

$$= I(U_{1:|L|+1}|Y). \tag{20}$$

Since $I(U_{1:|L|+1}|Y)$ is non-negative, *Conditional Redundancy* monotonically increases.

*Redundancy* is total correlation (multi-information). We can similarly consider its increase when $U_{1:|L|}$ obtains a new element, $\mathbf{u}^{(|L|+1)}$ as follows:

$$\mathcal{I}(U_{1:|L|+1}) - \mathcal{I}(U_{1:|L|}) \tag{21}$$

$$= \left(\sum_{i=1}^{|L|+1} H(\mathbf{u}^{(i)})\right) - H(U_{1:|L|+1}) - \left(\sum_{i=1}^{|L|} H(\mathbf{u}^{(i)})\right) + H(U_{1:|L|}) \tag{22}$$

$$= H(\mathbf{u}^{(|L|+1)}) - H(U_{1:|L|+1}) + H(U_{1:|L|}) \tag{23}$$

$$= H(\mathbf{u}^{(|L|+1)}) - \left(H(\mathbf{u}^{(|L|+1)}) + H(\mathbf{u}^{(|L|+1)}|U_{1:|L|})\right) + H(U_{1:|L|}) \tag{24}$$

$$= H(U_{1:|L|}) - H(\mathbf{u}^{(|L|+1)}|U_{1:|L|}) \tag{25}$$

$$= I(U_{1:|L|+1}). \tag{26}$$

Since $I(U_{1:|L|+1})$ is non-negative, *Redundancy* monotonically increases. Note that Studeny (2006) introduces the monotonicity of total correlation.

Regarding *Information Theoretic Diversity*, we can consider its increase when $U_{1:|L|}$ obtains a new element, $\mathbf{u}^{(|L|+1)}$ by utilizing Eqs. 20 and 26 as follows:

$$\mathcal{I}(U_{1:|L|+1}|Y) - \mathcal{I}(U_{1:|L|}|Y) - (\mathcal{I}(U_{1:|L|+1}) - \mathcal{I}(U_{1:|L|})) \tag{27}$$

$$= I(U_{1:|L|+1}|Y) - I(U_{1:|L|+1}). \tag{28}$$

When elements in $U_{1:|L|+1}$ are independent but not independent given $Y$, Eqs. 28 is non-negative. On the other hand, when elements in $U_{1:|L|+1}$ are not independent but independent given $Y$, Eq. 28 is non-positive. Thus, *Information Theoretic Diversity* does not have monotonicity.

Finally, Iyer et al. (2021) indicates the non-monotonicity of mutual information. Therefore, $I(U_{1:|L|};Y)$ does not satisfy the monotonicity. We can check its increase when $U_{1:|L|}$ obtains a new element, $\mathbf{u}^{(|L|+1)}$ as follows:

$$I(\mathbf{u}^{(|L|+1)};Y) + I(U_{1:|L|+1}|Y) - I(U_{1:|L|+1}). \tag{29}$$

When elements in $U_{1:|L|+1}$ are independent but not independent given $Y$, Eq. 29 is non-negative. On the other hand, when elements in $U_{1:|L|+1}$ are not independent but independent given $Y$, Eq. 29 is non-positive.

### A.6 THE PROOF FOR THEOREM 6

Only $I(U_{1:|L|};Y)$ is a term including $U_{1:|L|}$ in both bounds, $I(U_{1:|L|};Y)$ dominates the bounds. Because $I(U_{1:|L|};Y)$ is not a monotonic function, the upper and lower bounds do not satisfy the monotonicity.

### A.7 THE PROOF FOR THEOREM 7

Because $I(U_{1:|L|};Y)$ is a joint mutual information, it is submodular and non-decreasing on $U_{1:|L|}$ under the conditional dependence of the variables in $U_{1:|L|}$ given $Y$ (Krause & Guestrin, 2005). To check that, we define the change of $I(U_{1:|L|};Y)$ when adding $\mathbf{u}$ to $U_{1:|L|}$ as follows:

$$\Delta(\mathbf{u}|U_{1:|L|}) = I(U_{1:|L|}, \mathbf{u}^{(|L|+1)};Y) - I(U_{1:|L|};Y) \tag{30}$$

$$= I(Y, \mathbf{u}|U_{1:|L|}) + I(U_{1:|L|};Y) - I(U_{1:|L|};Y) \tag{31}$$

$$= I(Y, \mathbf{u}|U_{1:|L|}) \tag{32}$$

Since $I(Y, \mathbf{u}|U_{1:|L|})$ is no less than zero, $I(U_{1:|L|};Y)$ monotonically increases when $U_{1:|L|}$ increases under the conditional dependence of the variables in $U_{1:|L|}$ given $Y$. By utilizing this reformulation,

$$I(Y, \mathbf{u}|U_{1:|L|}) = H(\mathbf{u}|U_{1:|L|}) - H(\mathbf{u}|Y, U_{1:|L|}), \tag{33}$$

Table 2: Dataset names and licenses.

| Dataset Name | License |
|---|---|
| AG News | CC BY 4.0 |
| ARC | CC BY-SA 4.0 |
| BoolQ | CC BY-SA 3.0 |
| MNLI | CC BY 4.0 |
| QQP | Apache 2.0 |
| RTE | Unknown |
| SST-2 | Unknown |
| WiC | CC BY-NC-SA 3.0 |

and utilizing entropy's submodularity, we can induce the following inequality:

$$\Delta(\mathbf{u}|U_{1:|L|}) - \Delta(\mathbf{u}|U_{1:|L|+1}) \tag{34}$$

$$=H(\mathbf{u}|U_{1:|L|}) - H(\mathbf{u}|Y,U_{1:|L|}) - \big(H(\mathbf{u}|U_{1:|L|+1}) - H(\mathbf{u}|Y,U_{1:|L|+1})\big) \tag{35}$$

$$=H(\mathbf{u}|U_{1:|L|}) - H(\mathbf{u}|U_{1:|L|+1}) - \big(H(\mathbf{u}|Y,U_{1:|L|}) - H(\mathbf{u}|Y,U_{1:|L|+1})\big) \tag{36}$$

$$=H(\mathbf{u}|U_{1:|L|}) - H(\mathbf{u}|U_{1:|L|+1}) - (H(\mathbf{u}|Y) - H(\mathbf{u}|Y)) \tag{37}$$

$$=H(\mathbf{u}|U_{1:|L|}) - H(\mathbf{u}|U_{1:|L|+1}) \tag{38}$$

$$\geq 0 \tag{39}$$

Therefore, $I(U_{1:|L|};Y)$ is submodular on $U_{1:|L|}$ under the conditional dependence of the variables in $U_{1:|L|}$ given $Y$.

*Information Theoretic Divergence* is a sum of total correlation and conditional total correlation. Both functions are supermodular according to Fujishige (1978); Studeny (2006). Since $\mathcal{I}(U_{1:|L|}|Y)$ is zero under the given condition, *Information Theoretic Diveristy* becomes

$$-\mathcal{I}(U_{1:|L|}). \tag{40}$$

Here $-\mathcal{I}(U_{1:|L|})$ is submodular and thus, *Information Theoretic Diveristy* is a submodular function on $U_{1:|L|}$.

### A.8  THE PROOF FOR THEOREM 8

Under the condition, since $I(U_{1:|L|};Y)$ is a submodular function, $-I(U_{1:|L|};Y)$ is a supermodular function. Only $-I(U_{1:|L|};Y)$ is a term including $U_{1:|L|}$ in both bounds. Therefore, the upper and lower bounds are supermodular functions and monotonically decrease.

## B  EXPERIMENTAL DETAILS

Table 2 shows the datasets and their licenses. Excluding the AG New dataset, we used a validation split because the test set has not been released. Table 3 shows the templates used for each task. All templates are extracted from promptsource (Bach et al., 2022)[6]. The output vocabulary is restricted to the tokens used in options corresponding to the task. Table 4 shows the details of the used models in our experiments. In the implementation, we modified the released code from Chang et al. (2024)[7]. We used an NVIDIA RTX A6000, which has 48GB of VRAM, to run the models.

## C  DETAILED RESULTS

Figure 7 shows the detailed results of the MSE, bias, diversity, and performance of LLMs for each layer.

---

[6]https://github.com/bigscience-workshop/promptsource
[7]https://github.com/terarachang/LLMDecomp

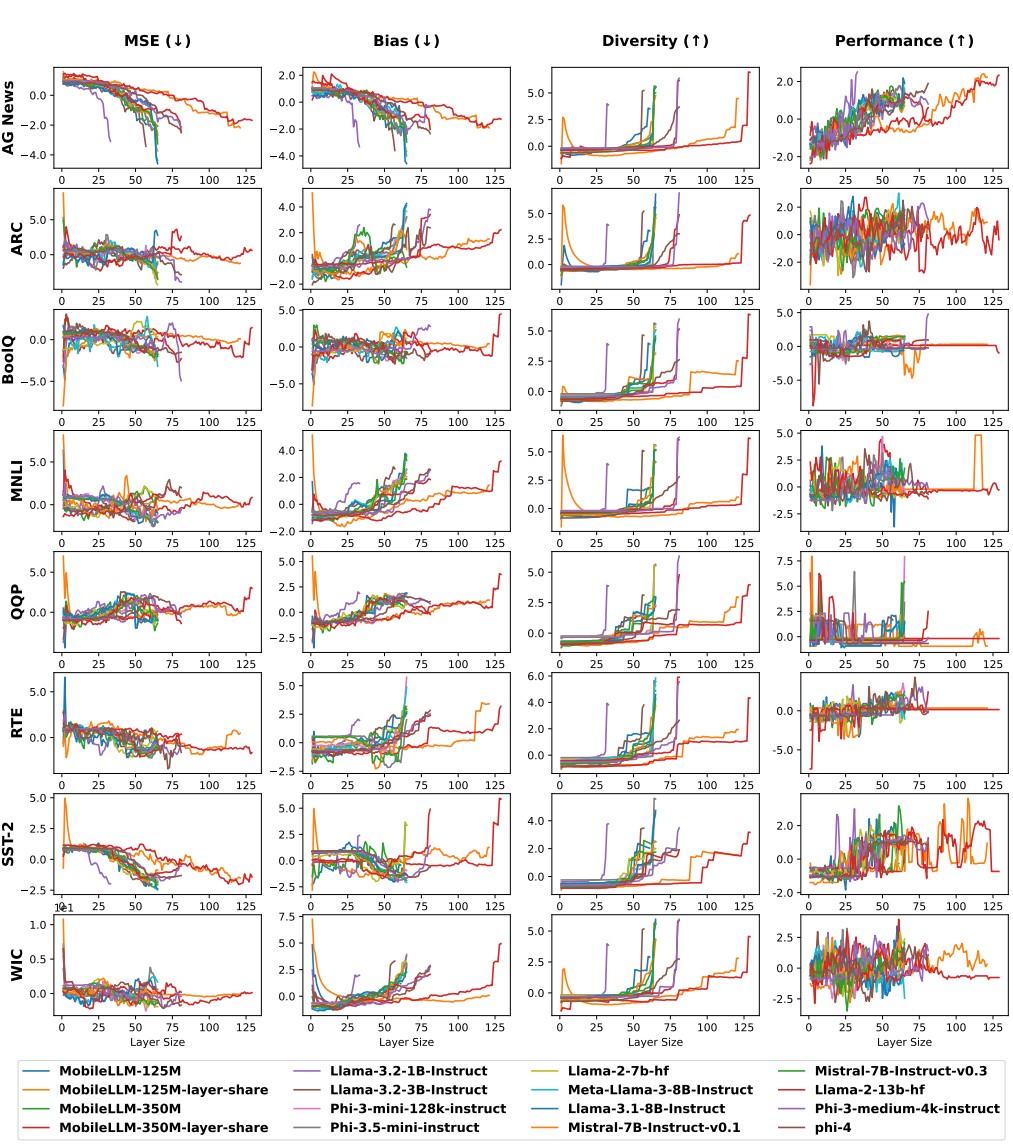

Figure 7: The relationship between MSE, bias, diversity, and performance of all models on each task. The settings are the same as Figure 5.

Table 3: The used templates for each task.

| Task | Template | Options |
|---|---|---|
| AG News | What label best describes this news article?\n {text} {answer} | World, Sports, Business, Science |
| ARC | Pick the most correct option to answer the following question.\n\n{text} \n\n{options} {answer} | A B C D |
| BoolQ | {passage}\n\nAfter reading this passage, I have a question: {question} True or False? {answer} | False, True |
| MNLI | Suppose it's true that {text1} Then, is "{text2}" {options} {answer} | Always, Sometimes, Never |
| QQP | I'm an administrator on the website Quora. There are two posts, one that asks "{question1}" and another that asks "{question2}" I can merge questions if they are asking the same thing. Can I merge these two questions? {answer} | no, yes |
| RTE | {text1} Using only the above description and what you know about the world, is {text2} definitely correct? Yes or No? {answer} | Yes, No |
| SST-2 | {text}\nQuestion: Was that sentence positive or negative? Answer: {answer} | negative, positive |
| WIC | Does the word "{word}" have the same meaning in these two sentences? Yes, No?\n {sentence1}\n{sentence2} {answer} | No, Yes |

Table 4: The statistics of the used models.

| Model Name | Parameter Size | Layer Size | License |
|---|---|---|---|
| facebook/MobileLLM-125M | 125M | 61 | MIT |
| facebook/MobileLLM-125M-layer-share | 125M | 121 | MIT |
| facebook/MobileLLM-350M | 350M | 65 | MIT |
| facebook/MobileLLM-350M-layer-share | 350M | 129 | MIT |
| meta-llama/Llama-3.2-1B-Instruct | 1B | 33 | LLaMA License |
| meta-llama/Llama-3.2-3B-Instruct | 3B | 57 | LLaMA License |
| microsoft/Phi-3-mini-128k-instruct | 3.8B | 65 | MIT |
| microsoft/Phi-3.5-mini-instruct | 3.8B | 65 | MIT |
| meta-llama/Llama-2-7b-hf | 7B | 65 | LLaMA License |
| mistralai/Mistral-7B-Instruct-v0.1 | 7B | 65 | Apache 2.0 |
| mistralai/Mistral-7B-Instruct-v0.3 | 7B | 65 | Apache 2.0 |
| meta-llama/Meta-Llama-3-8B-Instruct | 8B | 65 | LLaMA License |
| meta-llama/Llama-3.1-8B-Instruct | 8B | 65 | LLaMA License |
| meta-llama/Llama-2-13b-hf | 13B | 81 | LLaMA License |
| microsoft/Phi-3-medium-4k-instruct | 14B | 81 | MIT |
| microsoft/phi-4 | 14.7B | 81 | MIT |

## D    LLM USAGE

We used GPT-4o (OpenAI et al., 2024) only for proofreading our paper.

