# OpenReview forum: "Diversity of Transformer Layers: One Aspect of Parameter Scaling Laws"
_ICLR.cc/2026/Conference — Submitted to ICLR 2026_

### Official Review · Reviewer_Ufo2 · 2025-10-20

**Soundness:** 2
**Presentation:** 1
**Contribution:** 2
**Rating:** 2
**Confidence:** 4

**Summary:**

The work introduces a bias-diversity decomposition framework to theoretically analyze the contributions of individual layers (e.g., embedding, attention, MLP), aiming to explain some phenomena and outcomes of layer scaling.

**Strengths:**

>s1: Proposes a bias-diversity decomposition for analyzing layer-wise contributions in Transformer residual streams.

>s2: The work integrates the bias-diversity decomposition with an information-theoretic lens.

>s3: It has a theoretical and empirical analysis.

**Weaknesses:**

>w1: **There is a critical point of confusion and error in the technical details. Equation (1) implies that the outputs of the embedding layer, all MHA layers, and all MLP layers are summed together directly. This is incorrect.** For the correct sequential structure, please refer to [1].
>
> [1] On layer normalization in the transformer architecture. ICML 2020.

>w2: For Section 3.1. Omitting the softmax operation, while understandable for analytical convenience, requires explicit justification to ensure the theoretical rigor of the analysis.

>w3: Line 135-137. **Theorem 1, derived from Equation 1, requires further validation (See w1).** The authors should provide additional supporting assumptions and evidence to substantiate its theoretical soundness.

>w4: The formulation of Theorem 2 requires refinement for clarity. Please state its central claim and underlying rationale more explicitly.

>w5: I have concerns regarding Theorem 4. The application of the information-theoretic chain rule appears incomplete, as the dependencies among the variables {$u_i$} are not adequately accounted for in the derivation, especially as manifested in Equation (13).

>w6: **Line 250-252 "These results seem to indicate that". There are some gaps between the theory and the conclusions, requiring more details of the actual contributions. Instead of simply using expressions like "seem" to attempt an explanation.**

>w7: Section 5. The authors used relatively simple tasks such as SST-2 and RTE for models like llama3-8B, and more complex tasks are needed to validate their conclusions. For instance, in Lora+ [2], Roberta-base can already achieve over 0.9 accuracy on the SST-2 task.

>[2] LoRA+: Efficient Low Rank Adaptation of Large Models. ICML 2024.

**Questions:**

See Weaknesses. The authors provided an invalid anonymous link.

---

### Official Review · Reviewer_2UE7 · 2025-10-20

**Soundness:** 2
**Presentation:** 2
**Contribution:** 2
**Rating:** 4
**Confidence:** 4

**Summary:**

This paper investigates the role of layer-wise diversity in Transformers and its connection to parameter scaling laws. The authors propose a bias-diversity decomposition framework for the residual stream to theoretically and empirically analyze how the interplay between diversity and bias across layers and modules (e.g., attention, MLP) influences model performance. By introducing an information-theoretic perspective, they explain why increasing depth yields diminishing returns, aligning with the submodularity observed in scaling laws. Experiments on various LLMs and NLP benchmarks substantiate these theoretical findings.

**Strengths:**

1. A key strength of this work is its novel theoretical framework for bias-diversity decomposition, which it tailors specifically to the Transformer's residual stream. While grounded in ensemble learning, the formulation (Theorem 1, Eq. 8) provides an original and principled lens to quantify the contributions of individual layers and modules (e.g., attention, MLP) in granular, interpretable terms.
2. A particularly deep theoretical contribution is the reframing of diversity via information-theoretic measures (Eq. 13). This formalism allows the authors to establish a direct, causal link between layer stacking and submodularity, theoretically deriving—rather than just observing—the diminishing returns that characterize LLM scaling laws.
3. The experiments robustly substantiate the theory across a wide range of models and tasks. The evidence is compelling and multi-level: heatmaps (Fig. 3) establish macro-level correlations, module graphs (Fig. 4) offer microscopic insights into component behavior, and ablation figures (Fig. 5-7) meticulously track the interplay of key variables, collectively demonstrating the theory's robustness and generality.

**Weaknesses:**

1. The notation used throughout the paper is inconsistent and poorly standardized. Many symbols are not clearly defined upon their first appearance. The authors should provide a unified and clear definition for all symbols used in the theorems and mathematical formulae.
2. The results in Figures 3 and 5 do not strongly support the authors' viewpoint. In Figure 3, there is no clear correlation shown between bias, diversity, and accuracy, while the strong correlation between MSE and accuracy is not surprising. As for Figure 5, the changes in model performance across multiple tasks do not show a clear pattern.
3. Across all tasks shown in Figure 5, regardless of how performance changes, diversity consistently demonstrates a similar trend: it first remains unchanged and then suddenly increases. Moreover, in the few effective task groups, the rise in diversity clearly lags behind the decrease in bias. This phenomenon indicates that the authors' view on the trade-off between bias and diversity does not necessarily hold true.

**Questions:**

see weakness.

---

### Official Review · Reviewer_Joen · 2025-10-30

**Soundness:** 3
**Presentation:** 1
**Contribution:** 3
**Rating:** 2
**Confidence:** 3

**Summary:**

This paper investigates how the diversity among Transformer layers influences model performance and connects it to parameter scaling laws. Using a bias–diversity decomposition framework, the authors show that optimal performance arises when layers produce outputs that are both accurate (low bias) and mutually different (high diversity). The study also demonstrates that adding layers improves performance only when they contribute diverse behaviors, and that such gains diminish submodularly, consistent with scaling laws. Experiments on various LLMs and NLP tasks empirically confirm that diversity is a key factor driving performance improvements.

**Strengths:**

The paper introduces an original bias–diversity decomposition framework to connect internal Transformer dynamics with scaling laws, offering theoretical insight.

The work presents some theorems to formalize key relationships:

(1) the decomposition of prediction error into bias and diversity;

(2) the trade-off between bias and diversity;

(3) the submodularity of performance gains from adding layers

Comprehensive experiments across multiple LLMs (e.g., LLaMA, Phi, Mistral) and NLP benchmarks confirm the theoretical predictions

**Weaknesses:**

1. Theorems 5 and 6 indicate that increasing the number of model layers does not necessarily improve performance. This paper does not provide relevant experimental examples to illustrate this phenomenon. Both Conditional Redundancy and Redundancy increase monotonically, but their difference may still increase or decrease monotonically.

2. Theorem 7 assumes that when U_i are independent, the improvement of model performance is submodular. This paper doesn’t explain the rationality of the independence assumption for U_i. In practice, U_i should be correlated.

3.What conclusions can be drawn from Figure 3? The correlations between Diversity, Bias and accuracy vary significantly across tasks, which is inconsistent with the previous theoretical analysis (decomposition of Diversity and Bias).

4.The trade-off between Diversity and Bias means that reducing bias leads to a decrease in Diversity. In Figure 5, does "no performance improvement" refer to the ARC dataset, while "performance improvement" refers to the AG News dataset? The paper needs to provide more detailed explanations of this. Additionally, Section 5.4 mentions that bias does not tend to zero in practice, and this point also requires more detailed elaboration. Furthermore, Section 5.4 doesn’t explain when the Trade-off will fail and the reasons for such failure.


5.The experiments in Figure 6 are overly simple. Specifically, as the number of layers increases, performance first increases and then decreases, instead of ” the effectiveness of increasing layers actually decreases”. Additionally, more elaboration is required on why the conclusion that Conditional Redundancy is greater than 0 can be drawn.

**Questions:**

See the weakness above.

---

### Official Review · Reviewer_Q2CT · 2025-10-30

**Soundness:** 3
**Presentation:** 3
**Contribution:** 3
**Rating:** 4
**Confidence:** 3

**Summary:**

This paper bridges mechanistic interpretability and parameter scaling laws in pre-layer normalization Transformers by decomposing prediction error in the residual stream via a bias-diversity framework, where bias measures per-layer deviation from ground-truth logits and diversity captures inter-layer output differences, revealing that low bias + high diversity minimizes MSE

**Strengths:**

1. Introduces a bias–diversity decomposition framework to analyze Transformer layers, offering a novel theoretical lens linking internal mechanisms to scaling laws.
2. Bridges mechanistic interpretability and empirical scaling-law studies, a connection rarely explored.
3. Strong theoretical grounding with multiple theorems supported by rigorous proofs.
4. Empirical validation across diverse LLM families and NLP benchmarks strengthens claims.
5. The paper is clearly structured, with smooth transitions between theoretical and empirical sections.

**Weaknesses:**

1. While the theory links diversity to performance, concrete architectural or training recommendations (e.g., diversity regularization or pruning criteria) are missing. Demonstrating such interventions could make the work more actionable.
2. Experiments focus only on NLP benchmarks. Testing on multimodal or vision Transformers could demonstrate broader applicability and confirm whether the diversity–scaling relationship generalizes beyond text models.
3. Although the work aims to connect interpretability and scaling laws, the link remains largely conceptual. More explicit layer-wise interpretability analyses (e.g., probing or attribution) could better ground the theoretical findings in observable model behavior.
4. the link of the code is invalid.

**Questions:**

1. Could the authors elaborate on how their information-theoretic diversity measure relates to previously used diversity notions in ensemble learning or representation learning (e.g., orthogonality, mutual information between features)?
2. The theory assumes that the number of layers is proportional to parameter count and that layers are independent given the output label. How sensitive are the results to violations of these assumptions, for example, in shared-layer or MoE architectures?
3. Could the authors discuss potential architectural or training interventions (e.g., promoting inter-layer diversity or penalizing redundancy) that might emerge from this framework? Would such interventions alter scaling behavior in practice?
4. is the same bias–diversity scaling relationship expected to hold in vision or multimodal Transformers, where layer specialization and redundancy differ? Any preliminary evidence or reasoning would be helpful.

---

### Meta-Review · Area_Chair_5BjF · 2026-01-05

**Summary:**

Several theorems are experimentally unsubstantiated and not tied to reasonable assumptions, and all reviewers pointed to multiple issues. The issue highlighted by the most reviewers was that the experiments do not substantiate a correlation between bias, diversity, and accuracy.

**Reviewer Concerns:**

No rebuttal N/A

**Reviewer Scores:**

No rebuttal, same scores.

---

### Decision · Program_Chairs · 2026-01-26

Reject